# Examination of in Factor V Leiden and Prothrombin II Thrombophilic Mutations in Czech Young Women Using ddPCR—Prevalence and Cost–Benefit Analysis

**DOI:** 10.3390/healthcare9121656

**Published:** 2021-11-29

**Authors:** Petra Riedlova, Dagmar Kramna, Silvie Ostrizkova, Hana Tomaskova, Vitezslav Jirik

**Affiliations:** 1Centre for Epidemiological Research, Faculty of Medicine, University of Ostrava, 70300 Ostrava, Czech Republic; dagmar.kramna@osu.cz (D.K.); silvie.ostrizkova@osu.cz (S.O.); hana.tomaskova@osu.cz (H.T.); vitezslav.jirik@osu.cz (V.J.); 2Department of Epidemiology and Public Health, Faculty of Medicine, University of Ostrava, 70300 Ostrava, Czech Republic

**Keywords:** thrombophilic mutation, factor V Leiden, factor II prothrombin, ddPCR, oral contraceptives, smoking

## Abstract

Background: Thrombophilic mutations in genes for factor V Leiden and factor II prothrombin are among the most important risk factors for developing the thromboembolic disease (TED), along with the use of oral contraceptives (OCs) or smoking. Aim: This study aimed to investigate the occurrence of risk factors in young women using droplet digital PCR (ddPCR) and, based on the results of this investigation, to perform a cost–benefit analysis of ddPCR-based screening in young women starting to take OCs compared to the treatment costs of patients who develop preventable TED in the Czech Republic. Methods: In this cross-sectional study, female university students filled in a questionnaire and provided a blood sample for DNA isolation and ddPCR analysis of both aforementioned genetic risk factors. The results, along with data from literature and web search, were used for cost–benefit analysis valid for the Czech Republic. Results: Out of 148 participants, 30 (20%) were smokers and 49 (33%) took OCs. A mutation was confirmed in 6 women (4.1%) in the factor V gene and in 3 women (2%) in the factor II gene, respectively. A model calculation on a cohort of 50,000 women starting to use contraceptives in the Czech Republic every year showed that at maximum compliance, (i.e., non-use of OC and smoking cessation), screening could prevent 68 cases of TED over the course of the mean period of OC use (5.7 years). Economically, the costs of testing in this cohort (2.25 mil. USD) would be significantly lower than prevented treatment costs (16 mil. USD at maximum compliance); the cost–benefit break-even point would be at 14.1% compliance. Conclusion: The cost–benefit analysis based on our results indicates that screening for factor V Leiden and factor II prothrombin in young women before starting to use OCs would, in the conditions of the Czech Republic, likely be highly economically effective.

## 1. Introduction

The presented study is a small part of the HAIE research, which, in addition to comparing the health impacts in an environmentally polluted industrial and non-industrial region, assesses health risks and their perceptions for different lifestyles of all ages. This article emphasizes the role of prevention and the responsible approach of young women to the health risks of arterial embolism and thrombosis (below the thromboembolic disease-TED, i.e., I 74 according to ICD-10) associated with the use of oral contraceptives and smoking [1]. TED is a common and serious disease, arising as a result of the formation of a blood clot (thrombus) in the vascular bed. The two most important life-threatening conditions counting among TED events are deep vein thrombosis (DVT) and pulmonary embolism (PE) [2,3,4,5,6,7,8], affecting approximately 2 out of 10,000 population annually [9]. TED can lead to other conditions with significant morbidity and mortality including post-thrombotic syndrome, extension of thrombi, pulmonary hypertension, and recurrence [2]. It can also result in stroke, myocardial infarction (IM), or problems during pregnancy. Risk factors include genetic predispositions, acquired alterations in the mechanism of clotting, or interaction between such factors. Of the genetic factors, two thrombophilic mutations in the genes for factor V Leiden (FV Leiden, Leiden mutation, G1691A) and factor II prothrombin (FII prothrombin, G20210A) participating in the process of blood clotting, are the most important [4,5,6,7,8,10]. FV Leiden is the most common congenital form of thrombophilia, representing approximately 40–50% of all thrombophilia cases. In the Czech population, it is estimated that there are 5–10% of carriers of FV Leiden and 1–2% of FII prothrombin mutations, respectively. The presence of these greatly increases (up to 100× at the homozygous form) [11] the risk of the formation of a thrombus and vessel obstruction. This risk synergistically increases further with the use of oral contraceptives (OCs) and smoking. The use of OCs is the main reason for the fact that young women are a group with an increased risk of TED [3,4,5,6,7,8,12].

OCs are used by over 100 million women worldwide. Nowadays, contraceptives with a low estrogen content are available; the increased risk of DVT associated with OC use, nevertheless, persists. The risk is highest in the first year of OCs use and, if other risk factors, such as congenital thrombophilia or smoking, are present, it increases further [7,13,14].

Individuals with mutations in FV and FII genes have no symptoms, and the presence of the mutation is usually revealed only as a result of a DVT or PE event. At present, routine screening of women before their first OCs prescription is not recommended, and the genetic analysis is performed only in individuals with events of: recurrent DVT, first unprovoked DVT, DVT in unusual locations, DVT associated with OC use, DVT during pregnancy or puerperium, recurrent abortions in the 1st trimester or stillbirth and DVT present in a family member at the age of 50 or younger.

These recommendations lead to performing genetic analysis only in a limited group of individuals (and, moreover, mostly only after adverse events have already occurred) and is practically in opposition to the conclusions of studies warning from the use of OCs in women with congenital thrombophilic mutations [7,15]. We hypothesize that the analysis of these two thrombophilic mutations in young women before their first use of OCs would be beneficial. In the Czech Republic, this would constitute approximately. 50,000 women every year [7,16,17]. Women in possession of the information that they have a thrombophilic mutation and, as such, have an increased risk of TED, could possibly induce a positive change in their lifestyles, especially by opting for other methods of contraception and, where applicable, by smoking cessation. Such an analysis would be performed once in a lifetime only and could also be beneficial in the course of their future life (e.g., if undergoing surgery, long-term immobilization, or treatment of chronic diseases) [7,16]. The economy of such screening in comparison with a possible benefit associated with preventing TED events, however, remains questionable.

Nowadays, nevertheless, the gradual progress in laboratory DNA analyses has brought methods that are suitable for screening purposes, such as relatively cheap (droplet-digital) PCR techniques that could lead to a reduction of the screening costs. Hence, we have performed the presented pilot study in young women, aiming (i) to find out the prevalence of risk factors (i.e., two thrombophilic mutations in FV and FII genes, use of OCs, and smoking) in young Czech women; (ii) to evaluate the suitability and costs of mutation screening using the novel droplet digital PCR (ddPCR) method, which appears to be a cheaper, while highly sensitive (0.001%), approach allowing absolute quantification [18]; based on these results, (iii) to determine the impact of the knowledge about the presence of thrombophilic mutation on risky behavior (OC use, smoking) and (iv) to perform a cost–benefit analysis of screening for the presence of a thrombophilic mutation in women planning to start using OCs in comparison with costs of treatment of preventable TED cases using figures valid for the Czech Republic.

## 2. Methods

This cross-sectional epidemiological study was performed at the Faculty of Medicine, University of Ostrava. Participation in the study was offered to female students of the University of Ostrava, of whom 148 volunteered. The principal motivation for participants was the acquisition of the information about their own health risks associated with thrombophilia and they were informed about the results of their individual examinations.

The study, as well as the informed consent that was signed by all participants, was approved by the Ethics Committee of the Faculty of Medicine, University of Ostrava. The participants also filled in a questionnaire containing questions regarding their personal and family history, as well as the presence of risk factors of TED (OCs, smoking). Subsequently, 2.5 mL of full blood were taken from each participant into Vacutainer tubes with EDTA, left for at least 2 h at room temperature and subsequently frozen to −24°C until the time of DNA isolation. This was performed manually using the NucleoSpin blood kit (Macherey-Nagel, Dueren, Germany) and the DNA concentration and purity were verified using a microvolume spectrophotometer (DeNovix, Wilmington, DE, USA). The DNA concentration was decisive for the amount of the DNA template used for subsequent ddPCR analysis. The reaction mixture contained the template, PCR water, ddPCR supermix (Biorad, Berkeley, CA, USA), and ddPCR mutation detection assay for the FV and FII mutations (Biorad, Berkeley, CA, USA), respectively.

Commercial prices of the analyses of individual thrombophilic mutations were acquired from the laboratory pricelists [12,19,20,21], costs of TED treatment in the Czech Republic were taken from Havlín et al. [17]. Individual risk factors (RF, risk ratio, RR) for TED development used for calculations were taken from studies from the Czech Republic or Central Europe. Where several studies reported various values, mean values were used for calculation [21,22,23,24,25].

The calculations of the numbers of cases in the individual subgroups were performed according to the formula:N_i_= P × I × f × RF_i_/10,000(1)
where

N_i_ = number of events in the individual subgroup per yearP = the considered study population (number of women starting to take OC per year; in our case, 50,000)I = incidence per 10,000 general population (in our case, 2 per 10,000 population per year)f = fraction of the total population with the particular risk factorRF_i_ = risk factor of TED occurring (relative to the general population) in the respective subgroup10,000 = adjustment to the incidence

The total number of prevented events at 100% compliance was then calculated as
N = T × (ΣN_i_ − N_non-preventable_)(2)
where

T is the mean period of OC use in the Czech Republicand N_non-preventable_ is the number of non-preventable cases (describing the risk in mutation carriers that cannot be affected by responsible behavior).

## 3. Statistical Evaluation

Basic descriptive statistics (arithmetic mean, standard deviation) were used for the characterization of the study group. The occurrence of the factors was expressed as the frequency with a binomial confidence interval (CI). All analyses were performed in Stata version 14 (StataCorp, College Station, TX, USA).

## 4. Results

### 4.1. Analysis of the Risk Factors

148 women aged 18 to 37 were included in the study, of which 30 (20%) were smokers and 49 (33%) took OCs. DVT and embolism were present in the family history of 10 and 3 women, respectively. One participant had a personal history of thrombosis. Two women in the patient group had a personal history of abortion and one of premature birth (6th month).

DNA concentrations in the purified samples ranged from 3.47 to 88.232 ng/µL, purity from 1.370 to 2.250. ddPCR detected factor V Leiden mutation in 6 women (4.1%, 95% CI: 1.5–8.6%), and mutation in the FII prothrombin gene in three women (2%, 95% CI: 0.42–5.8%). Two out of these women with a positive finding of thrombophilic mutation used OCs (22.2%) and two were smokers (22.2%). Hence, at least one of the studied mutations was present in 6.1% of women in our study group (95% CI: 2.8–11.2%); 44.4% of these had an additional risk factor (OCs, smoking) further increasing the risk of TED. One of them had a family history of embolism, one of myocardial infarction, and one had thrombosis in her personal history. The overview of the participants of the study, participants with thrombophilic mutation, and other risk factors, is detailed in Table 1.

All women in whom the mutation was detected were asked after 6 months if their lifestyle has changed from the perspective of the additional risk factors (OCs, smoking). Both women with the mutation who were using OCs before the analysis discontinued OCs and both women who smoked ceased smoking after receiving the information; the compliance in our sample was, therefore, 100%.

### 4.2. Cost–Benefit Analysis

The combined commercial price of the analysis of FV and FII mutations in the Czech Republic for self-payers utilizing current methods ranges from 56 to 140 USD [12,19,20,21].

The cost of ddPCR analysis of both mutations (reaction only, without sample isolation, labor, electricity, and instrument depreciation) was 9 USD if each sample was analyzed separately. However, if using the advantage posed by ddPCR, i.e., its capability to analyze two samples at the same time or two mutations in one sample, the reaction price drops to less than 7 USD. Most commercial laboratories at present use RT-PCR with a reaction price of approximately 18 USD (both mutations, only reaction price, as in the case of ddPCR). Where sequencers are used, the price is even higher. Nevertheless, if ddPCR was used and its capability of analyzing both mutations in a single properly utilized (especially in combination with an increased volume of analyses that would go hand in hand with the implementation of a screening program), it is possible to conservatively expect the price reduction compared to the current lowest prices by about 20%. The costs of the analysis of the two most common thrombophilic mutations (FV + FII) in the Czech Republic would, therefore, amount to approximately 45 USD. Considering the fact that every year approximately 50,000 women start to use contraceptives in the Czech Republic [17], the costs of screening of this cohort prior to the first prescription of contraceptives (i.e., of women who are planning to introduce a new TED risk factor in their lives) would amount to approximately 2.25 mil. USD.

Taking into account the prevalence detected in our patient group, i.e., should the thrombophilic mutation be found in 6.1% of women, it would be 3050 women out of this cohort of 50,000 (of which 2050 with FV, 1000 with FII) who would receive a recommendation to opt for other methods of contraception. Of these, at the prevalence of smokers of approximately 20%, 610 individuals would be recommended to cease smoking as well.

The risk of a TED event is up to 35× higher (mean value among studies 27.5×) than in the general population in women using OCs and up to 100× (mean 90×) higher in FV Leiden heterozygotes and homozygotes using OC, respectively [22,23]. As the FV Leiden heterozygote:homozygote ratio in the Czech population is approximately 30:1 [24], we can calculate with a weighted mean of the risk factor being 29.5. The risk in women using OC with FII mutation is 16x increased compared to the general population [22].

In women who would, in addition to the above, be also smokers (610 women in the analyzed cohort of 50,000), the risk would be further synergistically increased, resulting in a 31× increased risk and 17.5× increased risk of a TED event compared to the general population for the 400 women with FV and 210 women with FII mutations, respectively. The individual risks and annual incidence of TED events in the considered cohort including literature sources are detailed in Table 2.

Calculations detailed in Table 2 thus show that at a 100% compliance of patients with thrombophilic mutations (i.e., if these patients would not start taking OCs and ceased smoking), 12 TED events could be prevented annually in the cohort of 50,000 women. The mean length of using OCs in the Czech Republic is 5.7 years [27]; at a 100% compliance, therefore, we could expect prevention of 68.4 TED events in a single cohort of 50,000 women who start taking OCs in any particular year over the mean period of OCs use. Average lifelong costs of TED treatment in the Czech Republic are approximately 234,000 USD per patient [17]. Hence, the costs of lifelong treatment of these 68.4 prevented cases would be 16 mil. USD. After deducting the screening costs, the estimated (purely economical) savings would be 13.8 mil. USD.

Besides the cost–benefit analysis at 100% compliance (i.e., compliance observed in our study), we also calculated the savings at 80%, 50%, and 20% compliance and the break-even point (i.e., the compliance at which the screening costs are higher than prevented treatment costs), which was determined at 14.1% compliance. These values are presented in Table 3.

## 5. Discussion

The aims of our study were to (i) determine the prevalence of the risk factors for TED in young women in the Czech Republic and (ii) on this sample, to evaluate the feasibility and cost-effectiveness of mutation screening with current state-of-the-art methods using the ddPCR instrument. Based on these results, we aimed (iii) to determine the impact of the knowledge about the presence of thrombophilic mutation on risky behavior (OCs use, smoking) and (iv) to perform a cost–benefit analysis of screening for the presence of thrombophilic mutations in women planning to start the use of OCs compared to the treatment costs of TED patients (valid for the Czech Republic).

Up to 10% of women in the Czech Republic (the prevalence in our study was 6.1%) have one of the two most common thrombophilic mutations [28,29]. Improving prevention of TED events in these women is, therefore, highly desirable as 10% of the female population can thus be forewarned of health risks, and, for the cost of a once-in-a-lifetime analysis, this information brings out a potential to prevent such an event.

In our study, there were 9 women with thrombophilic mutations (6.1%), two of whom were taking OC and two of whom were smokers. After being informed of the result, all four eliminated these undesirable risk factors (stopped taking OC and smoking) and generally started to address their lifestyle, knowing the consequences they may face in the future. Moreover, neither of the women who have had no additional risk factors at the time of the study (i.e., have neither taken OCs OCs nor smoked) and were notified of the presence of thrombophilic mutations have started taking OCs or smoking since. The responsiveness to the information about thrombophilic mutations in our study was, therefore, 100%. However, this extremely high compliance may be caused by the population in our study, consisting entirely of female university students, who might be expected to have higher responsiveness than the general population. Such a high compliance, therefore, cannot be expected in the entire population, so we considered lower responsiveness in our calculations (see below).

33% of the women analyzed in our group were taking OCs without the knowledge of their status of thrombophilic mutations and without the knowledge that a combination of these risk factors can lead to a significant increase in their risk of TED events. As mentioned above, the current guidelines do not recommend routine genetic testing of women before starting taking OCs [7,15].

The ddPCR method proved to be highly sensitive and suitable for screening in our study, proving the capability of analyzing even the sample with the lowest DNA concentration (3.47 ng/µL). This low value was likely caused by the manual isolation of DNA; should automated isolation be used, the concentrations would likely be rather at the upper limit of the range of DNA concentrations acquired during DNA isolation in our study. Nevertheless, the presence of a sample with such a low concentration only verified that ddPCR is highly sensitive—no analysis had to be repeated due to the low DNA content. Still, the fact that ddPCR is capable of analyzing two samples (or two mutations within one sample) in one run and one well remains its greatest benefit compared to other PCR techniques, saving the costs of chemicals [18]. The method proved sensitive and relatively cheap, thus being highly suitable for screening young women due to the low costs. The reduction in the price of commercial analyses from 56 USD to 45 USD when implementing ddPCR and a high-volume screening program is, in our opinion, a rather conservative estimate.

Our results indicate that, at 100% compliance, screening of the cohort of 50,000 women starting to take OCs every year in the Czech Republic for the presence of the two most common thrombophilic mutations could prevent 12 TED events every year; considering the average period of OC use (5.7 years [27]), this amounts to 68.4 cases over the entire period of OC use. At the compliance observed in our study (100%), the purely financial benefit of screening of such a cohort would be 13.76 mil. USD over the time of their OC use. However, the economical effectiveness would be maintained even at lower compliances; the break-even point is the compliance (i.e., decision not to take OCs and/or smoking cessation as a result of the knowledge of the possession of thrombophilic mutation) of 14.1%. We assume that (especially in view of the fact that the greatest part of the risk is associated with OC and that it is at present relatively simple to opt for another method of contraception), the responsiveness would be likely well above this break-even point.

### Study Limitations

The presented study comes with limitations. The first limitation is a relatively small number of participants (148), which may have caused a relatively imprecise determination of the prevalence of thrombophilic mutations in the population. However, the result of our analysis (4.1% for Leiden V and 2% for prothrombin) is not far from prevalences reported elsewhere, although these were generally somewhat higher [2,3,4,5,6,7]. The prevalence detected in our study was, however, used for the cost–benefit analysis as we kept on the conservative side of the estimates.

The conservative approach was maintained also in the estimate of the possible target price using ddPCR under the circumstances of a massive screening program. Besides the reduction in costs of the analyses by approximately 20%, we should also take into account that over the course of the patients‘ lives, the genetic analysis is performed only once; this would make the later analysis of thrombophilic mutations, which would be likely performed in a relatively sizable portion of the population for various reasons, unnecessary. These costs should be, therefore, deducted from the screening costs (they would be incurred anyway, just at a later age in many cases). The knowledge of the presence already at a young age could possibly influence lifestyles beyond simple smoking cessation and OC avoidance, which could further benefit the healthcare system.

There is, however, a known downside to genetic screening—namely, the stress which may burden women with thrombophilic mutations and reduce their quality of life due to the fear of impending danger of a thrombophilic event [24]. However, despite these potentially negative effects on the psychology of some individuals, we still believe that preventing the potential development of the disease (especially as we are considering young women here) should take precedence. Although it is true that women with thrombophilic mutation might not develop TED over their whole lifetime and might feel unnecessarily stressed, improvement of their lifestyles is the the most likely consequence of such stress; this, in effect, may prevent the development of other diseases as well. For this reason, we perceive this proposed examination not negatively, but rather as a method providing the benefit of the knowledge of one‘s condition, which supports behavior preventing the disease development. Besides, we definitely do not suggest such testing to be compulsory; we demonstrate the economical and health benefits, thus arguing for this examination to be offered and paid for by national insurance. The decision whether to test or not to test would depend on the particular woman who would decide whether or not she wants this information (the calculation shows statistical proportions that would be valid also for smaller numbers).

In our study, we only calculated with data known for the Czech Republic and/or, in the case of risk factors, Central Europe, which is genetically relatively homogeneous. As the risk factors for women carrying mutation while using OC and smoking at the same time were not known, we used a simple additive calculation instead of synergistic action (thus, again, keeping at the conservative side of things). Risk factors for smokers were acquired from the study by Enga et al. [23]. In addition, we used simple arithmetic means where the results of several studies were available. This may be viewed as a limitation; however, calculating with upper and lower limits of confidence intervals would yield in combination extremely wide ranges of possible results.

Average lifelong costs of TED treatment in the Czech Republic are approximately 234,000 USD per patient [18]. This, however, does not take into account the age at which the patient developed TED. The real-life costs could, therefore, be higher with a longer life expectancy of young women. Besides, we have not taken into account the possible mortality resulting from TED events in young women (with the general value of life in the Czech Republic regardless of age being 865,000 USD; in young women, we might assume an even higher value of life) [30].

Besides, we have disregarded other risk factors for TED in our study, such as the environmental burden, which has been shown to also increase the risk of TED [31,32]. In the area of the Moravian-Silesian region where this project was performed, this can be a significant risk factor, as the area has been traditionally the heart of heavy industry in the Czech Republic. This can mean that in this area, the risk of TED events can be further increased and any reduction of any other risk factor can be particularly desirable. This association with environmental burden is another direction of the research presently investigated within the scope of our project (HAIE—healthy aging in industrial environment) [33].

According to our results, the introduction of screening of young women prior to the first prescription of OCs could make sense as even a rough economical analysis without the consideration of the social dimension or mortality of young women at the beginning of their productive age) shows a major economic benefit.

## 6. Conclusions

This pilot study proved to be highly useful for opening a discussion regarding the testing of thrombophilic mutations in young women planning to start taking OC. The presented cost–benefit analysis showed that such a screening would make sense (besides the populational and societal perspective—prevention of TED events, including possible deaths of young women) from the economical perspective as well; the financial benefit for the healthcare system would be maintained even if as few as 20% of patients in whom a thrombophilic mutation was detected avoided the use of OC.

## Figures and Tables

**Table 1 healthcare-09-01656-t001:** Presence of risk factors in the participants in the study.

**Number of participants**	148 (100%)	
**Thrombophilic mutations combined**	6.1% (95% CI: 2.8–11.2%)	
**Positively tested for Factor V Leiden**	6 (4.1%, 95% CI: 1.5–8.6%),	2 users of OCs (22.2%) and 2 smokers (22.2%)
**Positively tested for Factor II Prothrombin**	3 (2%, 95% CI: 0.42–5.8%).	
**OC users**	49 (33%)	
**Smokers**	30 (20%)	

FV—Factor V Leiden; FII—Factor II prothrombin.

**Table 2 healthcare-09-01656-t002:** Calculation of risk factors, incidences, and prevented cases at 100% compliance (i.e., if all women with a detected thrombophilic mutation restrain from taking OCs and cease smoking).

		Risk Factors (RR)	Calculation If the Number of Cases	Weighted Mean of Cases *	Risk Factors (RR)	Calculation of the Number of Cases	Weighted Mean of Cases *	Sum (Weighted Mean of Cases) FV + FII
Risk of TED occurrence	General population	1	Incidence in the general population (2/10,000) * risk factor * adjustment of the incidence to the number of people with Leiden (2050/10,000) *	0.41	1	Incidence in the general population (2/10,000) * risk factor * adjustment of the incidence to the number of people with Protrombin (1000/10,000) *	0.2	0.61
	Presence of mutation	7 [21,23,26]	Incidence in 10,000 general population (2) * risk factor * adjustment of the incidence to the number of people with Leiden (2050/10,000) *	2.87	3 [21,23,26]	Incidence in 10,000 general population (2) * risk factor * adjustment of the incidence to the number of people with Protrombin (1000/10,000) *	0.6	3.47
	Mutation + OC (non-smokers)	29.5 [22,23,26] *	risk factor * incidence in 10,000 general population (2) * adjustment to the number of non-smokers (1640/10,000)	9.68	16 [22]	risk factor * incidence in 10,000 general population (2) * adjustment to the number of non-smokers (800/10,000)	2.56	12.24
	Mutation + OC (smokers)	31	risk factor * incidence in 10,000 general population (2) * adjustment to the number of smokers (410/10,000)	2.54	17.5	risk factor * incidence in 10,000 general population (2) * adjustment to the number of smokers (200/10,000)	0.7	3.24
Number of cases prevented per year				9.34			2.66	12.01

* calculated as weighted mean from the homo/heterozygote data [24,25,26].

**Table 3 healthcare-09-01656-t003:** Calculation of compliance and the number of prevented cases in the cohort of 50,000 women starting to take OCs every year in the Czech Republic.

	Compliance				
	100%	80%	50%	20%	14.1%
Prevented cases/year	12	9.6	6	2.4	1.69
Prevented cases/5.7 years *	68.4	54.72	34.2	13.68	9.63
Prevented costs of treatment **	16 mil.	12.8 mil.	8 mil.	3.2 mil.	2.25 mil.
Screening costs per cohort (USD)	2.25 mil.				
Savings (USD)	13.8 mil.	10.6 mil.	5.76 mil.	0.95 mil.	0

* average period of taking contraceptives; ** Prevented costs of treatment of TED events occurring over the 5.7 years (USD).

## Data Availability

The data presented in this study are available on request from the corresponding author. The data are not publicly available due to restrictions privacy and ethical.

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
