# Peer review of "Examination of in Factor V Leiden and Prothrombin II Thrombophilic Mutations in Czech Young Women Using ddPCR—Prevalence and Cost–Benefit Analysis"

_healthcare, 2021, doi:10.3390/healthcare9121656_

Round 1

Reviewer 1 Report

I read with interest the study of Riedlova Petra et al. aiming to compare, through a cost-benefit analysis, the costs of screening for the presence of a thrombophilic mutations (i.e FV Leiden and prothrombin variant) in Czech Republic women planning to start using oral contraceptives vs. the costs of treatment of preventable thromboembolic disease. Considering a model calculation on a cohort of 50,000 women, the authors found a significant cost reduction for testing women for thrombophilia than for preventable thromboembolic disease. I found the paper well written. The results are clearly presented and the statistical analysis is solid. The main limitation, as reported by the authors in the paper, is the small sample size.

Here are my specific comments:

  • Another striking risk factor for thromboembolic disease, in addition to smoking and oral contraceptives as rightly reported in the paper by the authors, is obesity. It would be possible to add the effect of this risk factor in the cost-benefit analysis?
  • The references dealing with “thrombophilia” should be updated. In example, ref. n. 4 (Sadeghi-Hokmabadi, E et al.) is a minor study in this field and ref. n 7 (Kujovich, J. et al.) is an old one. According to FV Leiden I suggest to the authors to quote Campello E et al. Expert Rev Hematol. 2016 and as far as prothrombin variant is concerned Dziadosz M et al. Blood Coagul Fibrinolysis 2016. Check ref. n. 10 for accuracy.
  • I suggest to shortened the introduction and to remove the list reported at pag. 2 from line 72 to line 77.

Author Response

Response to Reviewer 1 Comments

Point 1: Another striking risk factor for thromboembolic disease, in addition to smoking and oral contraceptives as rightly reported in the paper by the authors, is obesity. It would be possible to add the effect of this risk factor in the cost-benefit analysis?

Response 1: This is a valid comment; Obesity was added to the introduction. However, we have not included it in our study for two reasons:

  1. its combined efffect with thrombophilic mutations is relatively lower than that of remaining considered factors
  2. we have based our calculations on the prevalences determined from our study population. As it consisted of university students (largely of the Faculty of Medicine) who are young and care for themselves, the number of obese individuals in the population would be grossly underestimated.

We included this in Limitations of the study.

Point 2: The references dealing with “thrombophilia” should be updated. In example, ref. n. 4 (Sadeghi-Hokmabadi, E et al.) is a minor study in this field and ref. n 7 (Kujovich, J. et al.) is an old one. According to FV Leiden I suggest to the authors to quote Campello E et al. Expert Rev Hematol. 2016 and as far as prothrombin variant is concerned Dziadosz M et al. Blood Coagul Fibrinolysis 2016. Check ref. n. 10 for accuracy.

Response 2: The study by Sadeghi-Hokmabadi et al. was replaced with Campello et al.  Diagnosis and management of factor V Leiden: Expert Review of Hematology: Vol 9, No 12 (tandfonline.com),

Review by Kujovich et al. was preserved – although older, we feel it would be unfair to omit it as we used it when writing the manuscript; nevererthless, the publicatrion by Dziadosz et al. was also used as suggested

Ref n. 10 by Simone et al. was incorrectly cited, thank you for that.

Changed in references.

Point 3: I suggest to shortened the introduction and to remove the list reported at pag. 2 from line 72 to line 77.

Response 3: I removed the bullets and put the individual points in one sentence.

Reviewer 2 Report

I suspect the authors have well over-estimated the beneficial impact of performing this analysis on all women on OC. They have not at all touched on the negatives of screening. This includes the fact that having an identifiable 'genetic mutation/disorder' has great negative psychological impact on affected individuals, as well as insurance implications and implications on family dynamics. Most women with FVL or PGM will not have a thrombosis, even if they are on OC's or smokers; so, to save those that may, a much greater number may need to modify lifestyle, potentially also suffer psychologically from a 'perceived risk' that may never come. Authors should balance their discussion highlighting the potential downside to a mass screening. As example, even at maximum compliance and saving of 68 cases of TED out of 50000 individuals, at a rate of 2-4% FVL/PGM, 1000-2000 individuals would receive notice of their genetic mutation and have to adjust/manage.

Table 2 was truncated in my pdf, so that the last column was not visible.

Minor - some grammatical and spelling issues, including 'Trombophilic' instead of 'Thrombophilic' in title. Myocardial infarction abbreviated as 'IM?

page 8: "Positive women who have not taken OCs or smoked at the time of the study have not started taking OCs or smoking." doesn't quite make sense.

Author Response

Response to Reviewer 2 Comments

Point 1: I suspect the authors have well over-estimated the beneficial impact of performing this analysis on all women on OC. They have not at all touched on the negatives of screening. This includes the fact that having an identifiable 'genetic mutation/disorder' has great negative psychological impact on affected individuals, as well as insurance implications and implications on family dynamics. Most women with FVL or PGM will not have a thrombosis, even if they are on OC's or smokers; so, to save those that may, a much greater number may need to modify lifestyle, potentially also suffer psychologically from a 'perceived risk' that may never come. Authors should balance their discussion highlighting the potential downside to a mass screening. As example, even at maximum compliance and saving of 68 cases of TED out of 50000 individuals, at a rate of 2-4% FVL/PGM, 1000-2000 individuals would receive notice of their genetic mutation and have to adjust/manage.

Response 1: This is true and thank you for the comment. We have added a paragraph pertaining to this issue into the Discussion.

It is necessary to say that we do not propose this as a compulsory screening but as an option that could be offered to the women and paid for by the state. Women who do not want to know this would not need to be tested, of course.

Point 2: Table 2 was truncated in my pdf, so that the last column was not visible.

Response 2: We apologize for this mistake by the system. In the submitteed manuscript, the whole table can be seen.

Point 3: Minor - some grammatical and spelling issues, including 'Trombophilic' instead of 'Thrombophilic' in title. Myocardial infarction abbreviated as 'IM?

Response 3: Trombophilic changed to Thrombophilic in the title. The abbreviation IM/MI was actually unnecessary and was removed from the text.

Point 4: Page 8: "Positive women who have not taken OCs or smoked at the time of the study have not started taking OCs or smoking." doesn't quite make sense.

Response 4: You are right, it was awkward. The sentence was amended for beter clarity.

Round 2

Reviewer 1 Report

No further comment

Reviewer 2 Report

no further suggestions